# Analysis of Estrogenic Activity in Maryland Coastal Bays Using the MCF-7 Cell Proliferation Assay

**DOI:** 10.3390/ijerph18126254

**Published:** 2021-06-09

**Authors:** Rehab Elfadul, Roman Jesien, Ahmed Elnabawi, Paulinus Chigbu, Ali Ishaque

**Affiliations:** 1Department of Natural Sciences, University of Maryland Eastern Shore, 1 Backbone Road, Princess Anne, MD 21853, USA; elfadul.rehab@epa.gov (R.E.); aelnabawi@umes.edu (A.E.); pchigbu@umes.edu (P.C.); 2Maryland Coastal Bays Program, 8219 Stephen Decatur Hwy, Berlin, MD 21811, USA; rjesien@mdcoastalbays.org

**Keywords:** cell proliferation, estrogenicity, MCF-7, agonist, antagonist ICI 182, 780, ER mediated

## Abstract

Contaminants of Emerging Concern (CECs) with estrogenic or estrogenic-like activity have been increasingly detected in aquatic environments and have been an issue of global concern due to their potential negative effects on wildlife and human health. This study used the MCF-7 cell proliferation assay (E-Screen) to assess the estrogenic activity profiles in Maryland Coastal Bays (MCBs), a eutrophic system of estuaries impacted by human activities. Estrogenic activity was observed in all study sites tested. Water samples from MCBs increased MCF-7 cell proliferation above the negative control from 2.1-fold at site 8, located in Sinepuxent Bay close to the Ocean City Inlet, to 6.3-fold at site 6, located in Newport Bay. The proliferative effects of the sediment samples over the negative control ranged from 1.9-fold at the Assateague Island National Seashore site to 7.7-fold at the Public Landing site. Moreover, elevated cell proliferation (*p* < 0.05) was observed when cells were co-exposed with 17ß-Estradiol (E2), while reduction in cell proliferation was observed when cells were co-exposed with the antagonist ICI 182, 780 suggesting that cell proliferative effects were primarily mediated by the estrogen receptor (ER). These results suggest the occurrence of some estrogenic or hormonal-like compounds in the MCBs and are consistent with our previous findings based on vitellogenin analyses.

## 1. Introduction

Contaminants of Emerging Concern (CECs), including estrogenic or estrogenic like compounds, have been an issue of global concern due to their potential negative effects on wildlife and human health [1,2]. Examples of classes of CECs are Pharmaceuticals and Personal Care Products (PPCPs), natural and synthetic hormones, and flame-retardants. Studies have shown the occurrence of these contaminants in environmental matrices such as water, sediment, and fish tissues in rivers, coastal waters, ground water, municipal wastewater, as well as drinking water [3,4,5,6,7,8,9].

The estrogenic activities of CECs have been confirmed through in vitro and in vivo studies. For example, phthalate esters were found to be estrogenic when tested in vitro [10,11,12]. Studies have also shown that UV filters are able to mimic naturally occurring hormones and substitute the estradiol in the estrogen receptor [10]. Soto et al. demonstrated that alkylphenols exhibit estrogenic activity when tested in vitro [11].

Exposure to environmental estrogens has been associated with a range of ecological impacts such as induction of vitellogenin in males and juveniles of fish, and causes intersexuality [13]. Several studies have provided a link between the increased rate of hormonal-dependent cancer and elevated levels of man-made chemicals, particularly the estrogen-like-compounds. While the role of Endocrine Disrupting Compounds (EDCs) in the elevation of breast cancer rates has not been clearly proven, several studies have shown the adverse effects of these emerging contaminants on breast cancer incidents due to their ability to mimic natural hormones and interact with their receptors [14]. As reviewed by Lecomte et al., over 70% of human breast cancer cases are estrogen dependent and they arise from ERα-positive [15]. Further, their growth and development are not only affected by endogenous estrogens, but also by environmental hormonal-like compounds. For example, the synthetic estrogen diethylstilbestrol (DES), which was used to treat some pregnant women from the 1950s to 1970s, has been linked to substantial fertility, birth-related effects and cancer to those women exposed in utero, in addition to their male and female offspring [16,17].

Matthiessen et al. demonstrated that the sources of estrogenic activities in streams could be attributed to the natural hormones E1 and E2 that are derived from livestock, phytoestrogens, in addition to human-derived hormones in septic tank overflows [13]. Studies have also confirmed the occurrence and distribution of estrogenic compounds in estuaries around the world. For example, Wang et al. demonstrated the existence of complex estrogenic pollution-risk gradients in Maluan Bay along the southeastern coast of China [18]. A study by do Nascimento et al. detected moderate estrogenic activity in Guanabara Bay in Southeastern Brazil [19]. Examples of some documented ranges of estrogenic activities in aquatic environments are shown in Table 1.

CECs vary extensively in their chemical structure and there is no defined common chemical structure accountable for their estrogenic activity. The wide structural variation of these compounds makes it difficult for a single chemical analysis to measure their fate and occurrence [25]. While chemical analysis is considered a significant tool for identification and quantification of compounds, the estrogenic potency of these detected compounds cannot be provided in these types of analyses. Some of the most significant advantages of using in vitro assays over chemical analysis are that mixture effects are taken into account [26]. A combination of chemical analysis with in vitro bioassays can be an effective method for investigating estrogen substances and can result in a complete description of the estrogenic activity of environmental complex mixture samples [26,27].

The MCF-7 human breast cancer cells are the prototype of ER-positive cells that are extensively used in both in vitro and in vivo studies as a model for estrogen-dependent cell proliferation [28,29]. The E-screen assay with MCF-7 cell lines is a sensitive and stable tool to quantitatively analyze environmental complex mixtures such as untreated wastewater extracts for their overall estrogenic activity [26]. In vitro studies have been successfully used to screen for estrogenic activity; for example, Picard et al. showed that N-butyl benzyl phthalate (BBP) stimulates MCF-7 cell proliferation in E-Screen assay, hence, it is considered a partial agonist [12]. According to Soto et al., xenoestrogen is considered a full agonist if it induces cell proliferation similar to the one achieved by 17ß-Estradiol (E2), while it is considered a partial agonist if it produces cell proliferation that is significantly lower than the one induced by E2, but at the same time higher than the negative control [11]. Soto et al. and Henry and Fair also described a method to test the anti-estrogenicity of the test compound by inducing the maximal cell proliferation of MCF7 cells with E2, followed by co-exposure of the cells with the test compound and measuring the decrease in proliferation, if it occurs [11,30]. This reduction in the proliferation rate indicates the anti-estrogenicity of the test compound.

Environmental samples can contain complex mixtures of both agonists and antagonists, and in vitro bioassays provide an estimate of the net effect of all estrogenic active compounds [25,31,32]. Subsequently, additional sample preparation, such as fractionation, might be required to indicate the influence of both agonist and antagonist [31].

### Description of the Study Location–Maryland Coastal Bays

Maryland Coastal Bays (MCBs) are shallow coastal lagoons located behind Ocean City, MD, USA and Assateague Island Seashore National Park with moderately uniform shallow depths. This system comprises of five major bays: Assawoman, Isle of Wight, Sinepuxent, Chincoteague, and Newport as well as the St. Martin River. The MCBs watershed covers an area of 175 mi^2^ in addition to over 117,000 acres of land, 71,000 acres of water, and 280 miles of coastline [33]. Like many other coastal zones around the US, MCBs area continues to experience rapid population growth and developmental pressure. Currently, MCBs are suffering some signs of strain from intense development, high nutrient and sediment loads, and other stresses associated with human activities [34]. For example, seagrasses, hard clams, and some fish populations are declining in the northern bays where there is high residential development. These signs of stress have begun to emerge in the southern bays, an area that used to be defined as pristine [35]. To our knowledge, there has never been a study on the estrogenic activity profiles of the MCBs. This study, therefore, aims to determine the estrogenic potency of water and sediment extracts from the MCBs. Figure 1 shows the map of the MCBs displaying the sampling sites, and Table 2 illustrates the description of the sampling sites.

## 2. Materials and Methods

### 2.1. Materials and Media

MCF-7 cell and Eagle’s Minimum Essential Medium with L-glutamine (EMEM) were purchased from American type culture collection ATCC (Manassas, VA, USA). Phenol red-free media (MEM α) was obtained from Gibco (Grand Island, NY, USA), charcoal-stripped fetal bovine serum and Penicillin Streptomycin were obtained from Fisher Scientific (Fair Lawn, NJ, USA). The antiestrogen fulvestrant (Faslodex, ICI 182, 780) was purchased from Tocris (Minneapolis, MN, USA). Ethanol, dimethyl sulfoxide (DMSO), 17βEstradiol (E2), and irgasan/triclosan (TCS) were purchased from Sigma–Aldrich (St Louis, MO, USA).

### 2.2. Sample Collection and Preparation

Surface water and sediment samples were collected in amber glass bottles from 13 sites in Maryland Coastal Bays (MCBs), in addition to influent and effluent samples from Princess Anne and Ocean Pines wastewater treatment plants. Samples were kept on ice until arrival at the laboratory. Water samples were filtered using GF/F 47 mm filter paper and sediment samples were dried using a lyophilizer. Estrogenic compounds were extracted from water and sediment samples by Solid Phase Extraction (SPE) using a modified Leuscha et al. procedure and EPA Method 3546, respectively [36,37]. Briefly, SPE was carried out for 2 L of water samples, using preconditioned Oasis HLB (SPE) cartridges at a flow rate of 10 mL/min while Microwave Assisted Extraction (MAE) was conducted on 5 g of sediment samples. The elution of the analytes of interest was carried out using methanol. The eluates were then evaporated to 1 mL portions using a nitrogen evaporator. Half of each eluate was evaporated to dryness using a gentle stream of nitrogen followed by the addition of 50 µL of DMSO. The volume of each extract was brought to 5 mL by using steroid-free experimental medium, and the solution was then sterile filtered through a 0.20 µm membrane filter. Seven dilutions were made from the stock solutions that contain 1% (*v*/*v*) DMSO.

### 2.3. Cell Culture

The MCF-7 human breast cancer cell lines (passage # 147) were obtained from the American Type Culture Collection (ATCC, Manassas, VA, USA) and stored in liquid nitrogen. Cells were routinely grown to 75–80% confluence in 75-cm^2^ flasks in EMEM supplemented with 5% FBS, 1% Penicillin Streptomycin and the cell cultures were maintained at 37 °C in a humidified incubator with a 5% CO_2_ atmosphere (following the commercial provider ATCC’s standard recommended conditions).

### 2.4. Study of the Estrogenicity of Three Common CECs

The estrogenic activity of three CECs that are commonly used including 17ß-Estradiol were examined in MCF-7 cell lines. 17ß-Estradiol dose–response curve was prepared by using concentrations that ranged between 10–11 M and 10–4 M. The EC50 values (50% inhibitory concentration) of 17ß-Estradiol, Acetaminophen, and Triclosan were measured using Nonlinear RegressioncDynamic Fitting (GraphPad Prism).

### 2.5. Cell Proliferation Assay (MTS)

The estrogenic activities of the water and sediment extracts were assessed by using MTS assay. This method was modified from the previously described method by Soto et al. [11]. In brief, MCF-7 cells were trypsinized and seeded into 96-well microtiter plates at an initial density of 104 cells per well. Cells were allowed to attach for 24 h; then, the medium (5% FBS supplemented to phenol red EMEM) was exchanged with experimental medium (5% charcoal stripped FBS supplemented to phenol red-free MEM). Cells were treated with seven dilutions of the organic extracts (1:1000 to 1:1,000,000), with three replicates for each dilution. To validate that the stimulation of cell proliferation was mediated by an ER mediated signal, the anti-estrogenicity of the organic extracts was measured using a method previously described by Soto et al. and Henry and Fair, in which the maximal proliferation of MCF-7 cells was induced using 17ß-Estradiol, followed by determining the reduction in cell proliferation that occurs as a result of co-exposure to the antiestrogen ICI 182, 780 [11,25]. After 5 days of exposure, the proliferation of the cells was measured by adding 20 μL of cell Titer MTS reagent and the plates were incubated at the previous conditions for 4 h. The absorbance was measured at 490 nm using a Synergy/2 microplate reader from Bio-Tek. All the experiments were carried out in duplicate.

### 2.6. Statistical Analysis

Results of cell proliferation were expressed as % of control, and all data were reported as mean ± SEM. Student’s t-test was used to determine the statistically significant differences on cell proliferation compared to negative control. Alpha (α) was set at 0.05 for all statistical tests and data with *p* ≤ 0.05, *p* ≤ 0.01, and *p* ≤ 0.005 were considered as significantly different, (*), (**), and (***), respectively.

## 3. Results

### 3.1. Assessment of the Estrogenicity of Three Common CECs

The estrogenic activity of 17ß-Estradiol, Triclosan, and Acetaminophen was measured in MCF-7 cell lines. Table 3 shows the classes and structures of the tested compounds and their calculated EC50 values. The estrogenic responses of MCF7-cells to the three chemicals are shown in Figure 2. 17ß-Estradiol induced the proliferation of the MCF7 cells by 4.7-fold, while Acetaminophen and Triclosan induced the proliferation of the cells to 2-fold compared to the negative control. The EC50 for the positive control (17ß-Estradiol) was 0.141 µM, while the EC50 for the antiseptic Triclosan and the pharmaceutical Acetaminophen were 0.033 and 2.41 µM, respectively.

### 3.2. Cell Proliferation Assay (MTS)

The estrogenic activities of the water and sediment extracts were assessed by using MTS assay and were reported as cell proliferation as a percent of negative control. Results are illustrated in Figure 3. For the water samples, the highest estrogenic activity was observed in site 6 (6.3-fold compare to the negative control) which is located in Newport Bay, while the lowest estrogenic activity was detected in site 8 (2.1-fold above the negative control), which is located in Sinepuxent Bay close to the Ocean City Inlet (Figure 3a). Untreated and treated extracts from Ocean Pines wastewater treatment plant induced MCF-7 cell proliferation by 4.7-fold and 3.4-fold, respectively, compared to the negative control, while the organic extract from a site immediately downstream of the plant induced MCF-7 cell proliferation by 4.2-fold compare to the negative control (Figure 3b).

OPWW = Ocean Pines Waste Water; PAWW = Princess Anne Waste Water

For the sediment samples, the highest estrogenic activity was induced by Public Landing organic extract (7.7-fold compare to the negative control) followed by Lewis Road Kayak Launch (4.2-fold compare to the negative control), while the lowest estrogenic activity was induced by Assateague Island National Seashore (1.9-fold above the negative control) (Figure 4a). Figure 4b showed MCF-7 cell proliferation of the 2013 water extracts collected from site 2, site 12, and site 13. Figure 5 shows that water and sediment extracts, extract plus 17ß-Estradiol, and extract plus antagonist ICI 182, 780 from a site immediately downstream from Ocean Pines wastewater treatment plant exhibited dose-dependent proliferation effects on MCF-7 cell lines.

## 4. Discussion

### 4.1. Study the Estrogenicity of Three Common CECs

The estrogenic activity of three commonly used CECs (17ß-Estradiol, Triclosan, and Acetaminophen) were examined. Picard et al., indicated that 17ß-Estradiol at concentrations between 10–12 M and 10–8 M stimulated MCF-7 cell proliferation in a concentration-dependent pattern [12]. Results from this experiment have shown that all three tested compounds significantly increased MCF-7 cell proliferation in a dose-dependent manner.

### 4.2. Cell Proliferation Assay (MTS)

In this study, the occurrence of CECs in MCBs was investigated by using MTS assay. The estrogenic activities of the water and sediment extracts were assessed and were reported as cell proliferation as a percent of negative control. The highest proliferative effect for water samples was observed in Newport Bay, while the lowest proliferative effect was detected in Sinepuxent Bay. The Public Landing site, followed by Lewis Road Kayak Launch, induced the highest proliferative impact in the sediment samples, while sediment from Assateague Island National Seashore induced the lowest proliferative effect.

The treated effluents from both PAWWTP and OPWWTP showed significantly high estrogenic activity close to the levels exhibited in the two influent samples, illustrating the insufficient removal of the estrogenic compounds during the treatment processes. According Verlicchi et al., common wastewater treatment plants are unable to remove CECs, hence, the documented higher estrogenic activity in both Ocean Pines and Princess Anne effluent samples are expected [42]. Furthermore, the organic extract from Ocean Pines wastewater treatment plant showed high estrogenic activity, although lower than predicted indicating the occurrence of anti-estrogenic substances, which suppressed the estrogenic activity. These results were consistent with a previous study conducted by Ihara et al., which illustrated that unfractionated wastewater extract induces weaker estrogenic activity compared to the fractioned one, indicating that anti-estrogenic compounds in wastewater defeat the action of estrogenic or estrogenic like compounds [43]. Moreover, the organic extract from the site immediately downstream of Ocean Pine WWTP induced MCF-7 cell proliferation close to the proliferation rate of the influent sample, confirming the finding by Verlicchi et al., which indicated that CECs are continuously released into the aquatic environments from WWTPs [42]. Our results indicate the anthropogenic contribution of Ocean Pine WWTP on MCBs receiving waters.

The measured estrogenic activity in AINS site was higher than predicted but in good agreement with the results from our vitellogenin analysis, which showed elevated levels of Vtg in male striped killifish and Mummichog occupying this site (unpublished). It should be noted that the distribution of wildlife in the ASP site could be associated with the estrogenic activity found there. Previous screening over multiple years suggests the presence of CECs in National Park waters in the semi-arid southwest [44].

Picard et al. indicated that the full estrogen agonist 17ß-Estradiol at concentrations between 10–12 M and 10–8 M stimulated MCF-7 cell proliferation in a concentration-dependent pattern [12]. A combination of the organic extracts with 17ß-Estradiol caused an additional increase in cell proliferation indicating that the extract has estrogenic compounds which enhanced the stimulation of the cell proliferation [45]. ICI 182,780 (Fulvestrant) is a pure anti-estrogen that binds to the ER and inhibits receptor dimerization, preventing nuclear localization of the receptor, as well as blocking and degrading the ER protein [46,47]. Addition of ICI 182, 780 to the extracts resulted in the decrease in the rate of proliferation for water and sediment samples, suggesting that the effects were mediated through the binding of the ER.

These in vitro results show that all water and sediment extracts from MCBs have the ability to induce proliferation of MCF-7 cells though to different extents compared to the control. When comparing the trend of the estrogenic activity for water samples among sites, site 6 (in Newport Bay) exhibited the highest estrogenic activity (63% over negative control) followed by OPUT (47% over negative control). Newport Bay is relatively poorly flushed, influenced by agricultural runoff, and received treated effluents from the Berlin Sewage Treatment Plant up to 2012 [48].

Additionally, when comparing the trend of the estrogenic activity for sediment samples among sites, the Public Landing site stimulated the highest MCF-7 cell proliferation (77% over negative control) followed by Lewis Road Kayak Launch (42% over negative control), Ocean Pines (37% over negative control), site 6 (28% over negative control), site 10 (21% over negative control), and Assateague Island National Seashore (19% over negative control). The high estrogenic activity in public landing site might be due to discharges from housing developments into Chincoteague Bay, or agricultural runoff. Also, it can be due to the leaching from the septic systems in the town of Chincoteague. According to Dennison et al., Chincoteague Bay watershed contains developmental, agricultural and forest areas in addition to the septic systems on sandy soil that can leak into the groundwater [44]. Significant cell proliferation was also observed with sediment extract from Lewis Road Kayak Launch site, an area used as landfill for Ocean City. This points to the significance of landfill leachates as a possible source of CEC to the ecosystems. According to Masoner et al., landfill leachates can contain complex mixtures of CECs that originate from a variety of sources [49]. A study conducted by Kawagoshi et al., detected estrogenic activity in leachates from a municipal waste-dumping site in Japan [50].

## 5. Conclusions

CECs with estrogenic or estrogenic-like activity have been increasingly detected in aquatic environments and have been an issue of global concern due to their potential negative effects on wildlife and human health. Several in vitro bioassays, including E-screen that measures the estrogenic activity of compound or compound mixtures on MCF-7 human breast cancer cell lines, have been used for screening the environmental hormonal-like compounds.

The aim of this study was to determine the estrogenic activity profiles of water and sediment extracts from MCBs. The estrogenic potencies of the extracts were examined by using MCF-7 cell proliferation assay.

Estrogenic activity was observed in almost all study sites tested (*p* < 0.05) including ASP, suggesting the occurrence of estrogenic or estrogenic-like compounds in MCBs. MCBs’ water samples increased MCF-7 cell proliferation between 2- to 5-fold above the negative control, while the proliferative effects of the sediment samples ranged between 2- to 8-fold over the negative control. Moreover, elevated cell proliferation (*p* < 0.05) was observed when cells were co-exposed with 17ß-Estradiol (E2) while reduction in cell proliferation was observed when cells were co-exposed with the antagonist ICI 182, 780, suggesting that cell proliferative effects were primarily mediated by the estrogen receptor (ER). Organic extract from Ocean Pines wastewater treatment plant showed higher estrogenic activity, but it was lower than predicted, indicating occurrence of anti-estrogenic substances, which suppress the estrogenic activity. To clarify this finding, further investigation of ocean pines extracts is required.

Results from E-Screen indicated the occurrence of biologically significant levels of estrogenic or estrogenic-like compounds in MCBs demonstrated by the elevated levels of MCF-7 cell proliferation compared to the negative control. This is well in agreement with previous findings based on biological analysis. Vitellogenin (Vtg) concentrations in male and juvenile fish were measured by using ELISA in order to assess the occurrence of CECs in MCBs. Results from this unpublished study showed that male Striped Killifish, Mummichog, and juvenile Atlantic Menhaden from MCBs were exposed to estrogenic or estrogenic-like compounds which was demonstrated by elevated levels of Vtg induction.

The estrogenic activities detected in this study may be due to the estrogenic or estrogenic-like compounds that were responsible for the Vtg detected in male and juvenile fish from the study sites.

## Figures and Tables

**Figure 1 ijerph-18-06254-f001:**
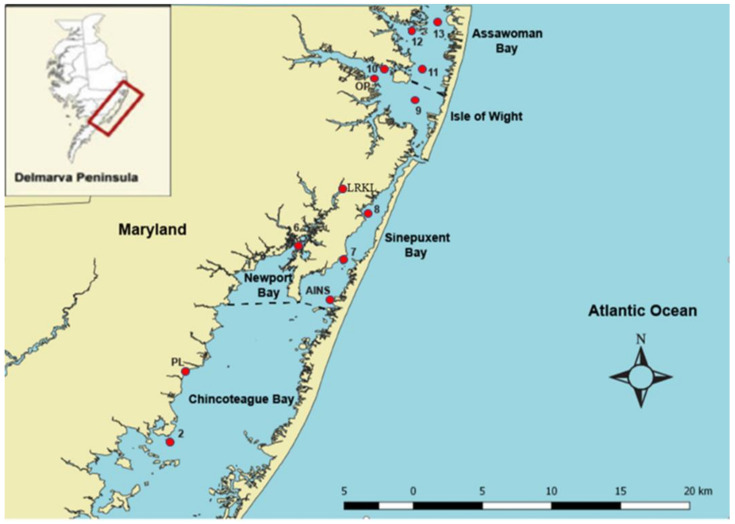
Map of the Maryland Coastal Bays showing the sampling sites.

**Figure 2 ijerph-18-06254-f002:**
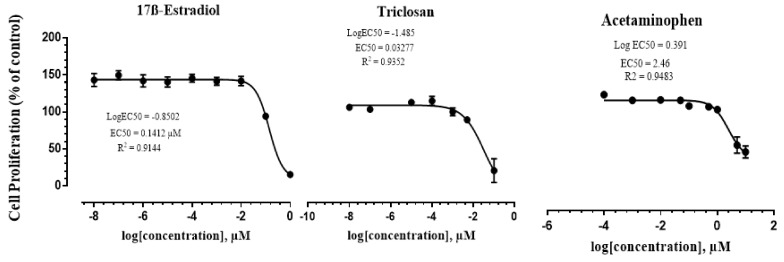
Dose–response curves of 17ß-Estradiol, Triclosan, and Acetaminophen on MCF7 cells after 72 h exposure when using MTS assay. Results are expressed as a percent of control. Data are presented as means ± SEM of two individual experiments. The EC50 values were calculated based on the dose response curve of each compound.

**Figure 3 ijerph-18-06254-f003:**
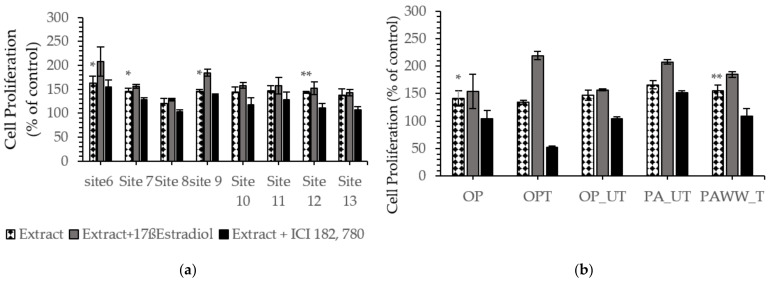
Effect of MCBs’ 2014 water extracts collected from (**a**) site 6, 7, 8, 9, 10,11,12,13; and from (**b**) OPWW and PAWW treated and untreated (1:10,000 to 1:1,000,000 dilution) on estrogen receptor MCF-7 cells proliferation after 5 days of exposure when using MTS assay. Results are expressed as a percent of control. Data are presented as means ± SEM of two individual experiments. Significant differences from negative control are represented as * *p* < 0.05, ** *p* < 0.01 and *** *p* < 0.005 respectively (Student *t*-test).

**Figure 4 ijerph-18-06254-f004:**
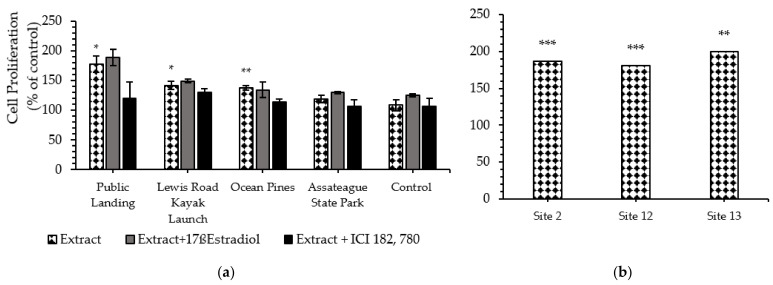
Effect of (**a**) 2013 sediment organic extracts collected from Public Landing, Lewis Road Kayak Launch, Ocean Pines, Assateague Island National Seashore and the negative control; and (**b**) 2013 water extracts collected from site 2, site 12, and site 13 (1:10,000 to 1:1,000,000 dilution) on estrogen receptor MCF-7 cells after 5 days of exposure when using MTS assay. Results are expressed as a percent of control. Data are presented as means ± SEM of two individual experiments. Significant differences from negative control are represented as * *p* < 0.05, ** *p* < 0.01 and *** *p* < 0.005 respectively (Student *t* test).

**Figure 5 ijerph-18-06254-f005:**
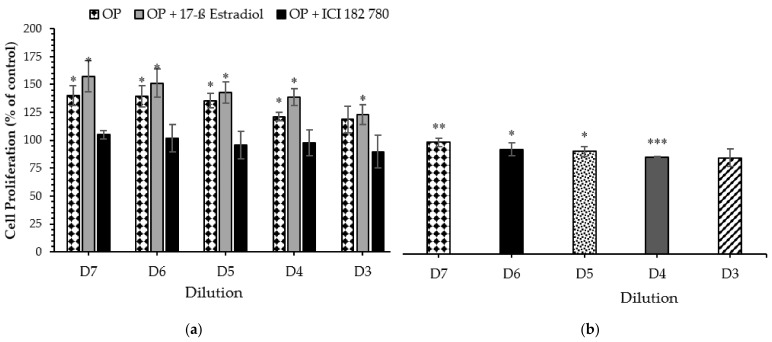
Dose-dependent Proliferation effect of OP: (**a**) water sample extract, extract + 17ß Estradiol and extract + antagonist ICI 182, 780; (**b**) Sediment sample extract (1:10,000 to 1:1,000,000 dilution) on estrogen receptor MCF-7 cells after 5 days of exposure when using MTS assay. Results are expressed as a percent of control. Data are presented as means ± SEM of two individual experiments. Significant differences from negative control are represented as * *p* < 0.05, ** *p* < 0.01 and *** *p* < 0.005 respectively (Student *t*-test).

**Table 1 ijerph-18-06254-t001:** Examples of studies that documented the range of estrogenic activities in the aquatic environment.

Aquatic Environment	Country	EEQ ^1^ (ng/L or ng/g)	Reference
Seawater	Japan	0.34–2.52	[20]
River water	Japan	0.70–4.01	[20]
Surface sediment	Japan	2.07–12.1	[20]
River water	China	8.15 (wet season)–34.7 (dry season)	[21]
Core sediment (River)	Italy	16.1 ± 9.3	[22]
Pore water (River)	Italy	20.5 ± 21.2	[22]
Bay water	Brazil	0.5–3.2	[19]
Bay surface water	Hong Kong	1.3	[23]
Bay Surface sediment	Hong Kong	5.9	[23]
Bay Surface Water	Germany	0.01–0.82	[24]

^1^ EEQ–17β-estradiol equivalents.

**Table 2 ijerph-18-06254-t002:** Description of the sampling sites.

Station No.	Station Name	Location	Lat	Long
Site 1 *	Stricking Marsh	Chincoteague Bay	38° 03.143	75° 16.114
Site 2	Assacorkin Island	Chincoteague Bay	38° 05.583	75° 17.860
Site 3 *	Pirate Island	Chincoteague Bay	38° 06.503	75° 13.369
Site 4 *	Public landing	Chincoteague Bay	38° 08.458	75° 16.051
Site 5 *	Chincoteague	Chincoteague Bay	38° 10.160	75° 13.909
Site 6	New port Bay	Newport Bay	38° 15.290	75° 11.593
Site 7	Sarbane Center	Sinepuxent Bay	38° 14.504	75° 09.306
Site 8	Duck blinds	Sinepuxent Bay	38° 16.825	75° 08.032
Site 9	The Flats	Isle of Wight Bay	38° 22.376	75° 05.736
Site 10	St. Martin River	Lower St. Martin River	38° 23.925	75° 07.302
Site 11	Light house sound	Assawoman Bay	38° 23.970	75° 05.428
Site 12	Grey’s creek	Assawoman Bay	38° 25.778	75° 05.956
Site 13	Fenwick Ditch	Assawoman Bay	38° 26.240	75° 04.651
OP	Ocean Pines	Ocean Pines Out flow	38.39086	−75.129339
OPWW_T	OPWW_T	Ocean Pines WW_Treated	N/A	N/A
OPWW_UT	OPWW_UT	Ocean Pines WW_Untreated	N/A	N/A
PAWW_UT	PAWW_UT	Princess Anne WW_Untreated	N/A	N/A
PAWW_T	PAWW_T	Princess Anne WW_Treated	N/A	N/A
AINS	Assateague Island National Seashore	Assateague Island National Seashore	38.210448	−75.16713
LRKL	Lewis Rd Kayak Launch	Lewis Rd Kayak Launch	38.15585	−75.15585
PL	Public Landing	Public Landing	38.152753	−75.283431

* Samples from sites 1, 3, 4 and 5 have not been analyzed.

**Table 3 ijerph-18-06254-t003:** Estrogenic response of MCF7 cells to 17ß-Estradiol, Acetaminophen, and Triclosan. The EC50 values were calculated based on the dose response curve of each compound.

Compound	Class	Structure	EC50 (µM)
17ßEstradaiol	Steroid [38]	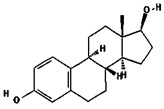 [39]	0.141
Acetaminophen	Pharmaceutical [40]	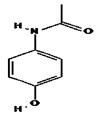 [39]	2.46
Triclosan	Antimicrobials [41]	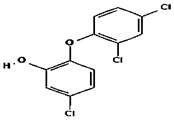 [39]	0.033

## Data Availability

The data are available on request.

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
