# Peer review of "Analysis of Estrogenic Activity in Maryland Coastal Bays Using the MCF-7 Cell Proliferation Assay"

_ijerph, 2021, doi:10.3390/ijerph18126254_

Round 1

Reviewer 1 Report

This study represents the experimental data and interpretation to aquatic environments on the estrogenic activity profiles in Maryland Coastal 14 Bays (MCBs). Particularly, authors were observed estrogenic activity by MCF-7 cell proliferative assay (E-screen) using sea water and its sedimented samples of each site.

In Table 1, as authors well know, almost sample were brought from river water or drinkable water in previous references. Authors maybe concern that natural circulation chain correlated to human being, public health and even specific disease. Whereas, this article has no sufficient clues including molecule and mechanism. Therefore, I think more reliable backgrounds and further research is needed.

Author Response

Responses to Reviewer 1 Comments

Point 1. In Table 1, as authors well know, almost sample were brought from river water or drinkable water in previous references

Response 1: More marine references have been added to Table 1 of the manuscript (p2, line 66) and to the reference section (p12, 438-442).

Point 2: Authors maybe concern that natural circulation chain correlated to human being, public health and even specific disease.

Response 2: The comments are not clear enough for us to provide a response.

Point 3: Whereas, this article has no sufficient clues including molecule and mechanism. Therefore, I think more reliable backgrounds and further research is needed.

Response 3: The focus of this research was not on the molecular mechanism of CFCs.

Reviewer 2 Report

The authors investigated about the effects of the estrogenic activity profiles in Maryland Coastal Bays using MCF-7 cells. Estrogenic activity was detected in all samples used in this study. When the cells were incubated with 17b-estradiol (E2), cell proliferation was further increased, but decreased when co-treated with estrogen receptor (ER) antagonist, ICI 182,780. Thus, observed estrogenic activity was occurred through ER. The results are sound. However, there are some concerns that should be addressed.

  1. Some figures are incomplete. Please confirm and they should be improved.
  2. Cell shape was changed by adding each extract. The cell morphology is shown or please add the comments to this concerns.
  3. Although “cell proliferation assay” mentioned in the title, the results are from cell viability assay. Time course-dependent cell growth should be measured by counting cell number, when each of these extracts are added in the culture.

Author Response

Response to Reviewer 2 Comment

Point 1: Some figures are incomplete. Please confirm and they should be improved.

Response 1: Figure 1 has been modified. Figure 5 (original)  and the associated findings have been removed. Figure A1 from the appendix has been transferred to the main text and renamed figure 5.

Point 2: Cell shape was changed by adding each extract. The cell morphology is shown or please add the comments to this concern.

Response 2: Cell morphology was not covered in this study but there was no change in morphology between the control and treated cells when viewed under the microscope.

Point 3: Although “cell proliferation assay” mentioned in the title, the results are from cell viability assay.

Response 3: Cell viability has been changed to cell proliferation in the text.

Point 4: Time course-dependent cell growth should be measured by counting cell number, when each of these extracts are added in the culture.

Response 4: Cell growth was measured by MTS assay which is more sensitive than counting. MTS assay is an indirect measure of cell number.

Reviewer 3 Report

The authors revealed endocrine disrupting disruption in specific areas. I request some minor improvements to this manuscript.

In the introduction, the authors elaborated on endocrine disrupters and screening assays, but both were only general basic knowledge. It is important to clarify the problems that the surveyed area has. And it should be clear what the authors wanted to reveal.

I couldn't understand the difference between the left and right sides of Figure 1.

In the results, the explanation in Figure 5 (p9, l246-248) was difficult to understand and needs to be improved.

In the discussion, the repeated description of the results was conspicuous. Also, the usefulness of the MCF7 screening assay has been established and may not be suitable for discussion.

The description of fulvestrant is incorrect (p10, l327-328). Fulvestrant inhibits ER dimerization and reduces ER protein. Improvements to accurate descriptions are needed.

Author Response

Response to Reviewer 3 Comments

Point 1:  In the introduction, the authors elaborated on endocrine disrupters and screening assays, but both were only general basic knowledge. It is important to clarify the problems that the surveyed area has. And it should be clear what the authors wanted to reveal.

Response 1:  Anthropogenic activities (highlighted on page 3, lines 100-113) have been taking place in the study area hence we were trying to access the level of their impact by using the level of contaminant of emerging concern (CEC) which can be a measure of anthropogenic activities.

Point 2: I couldn't understand the difference between the left and right sides of Figure 1.

Response 2: Figure 1 has been modified.

Point 3: In the results, the explanation in Figure 5 (p9, l246-248) was difficult to understand and needs to be improved.

Response 3: Figure 5 and the associated findings have been removed.  Figure A1 from the appendix has been transferred to the main text and renamed figure 5.

Point 4: In the discussion, the repeated description of the results was conspicuous. Also, the usefulness of the MCF7 screening assay has been established and may not be suitable for discussion.

Response 4: Most Results have been removed from the discussion and MCF7 usefulness as a screening assay has been removed from the discussion section.

Point 5: The description of fulvestrant is incorrect (p10, l327-328). Fulvestrant inhibits ER dimerization and reduces ER protein. Improvements to accurate descriptions are needed.

Response 5: Accurate description of fulvestrant has been provided (p9, 300-303).

Round 2

Reviewer 1 Report

  1. authors should cite references and defined structures to 17ß-Estradiol, Acetaminophen, and Triclosan in table 3.
  2. In table 3-5, why ICI 182,780 reduces more in combined group? is it not endogenous cytotoxicity? what dosages did use?
  3. this reviewer ask previously about mechanisms to this phenomenas. because this article suggested only effects of  ICI 182,780-mediated inhibition of estogenecity of materials in MCF-7 cell model. Please suggest the identities of the estrogenic effects of the materials in MCF-7 cells.

Author Response

Responses to Reviewer 1 Comments (Round2)

Point 1. authors should cite references and defined structures to 17ß-Estradiol, Acetaminophen, and Triclosan in table 3.

Response 1: Defined structures for 17ß-Estradiol, Acetaminophen, and Triclosan have been used and references have been added to Table 3 of the manuscript (p6, line 194) and to the reference section (p13, 475-483).

Point 2: In table 3-5, why ICI 182,780 reduces more in combined group? is it not endogenous cytotoxicity?

Response 2: Since there is no table 3-5, we believe the reviewer is referring to figure 3-5. It was anticipated that the combined extract and ICI 182,780 results would be lower than the control and the extract alone. Cell proliferation decreased in the ICI 182,780 and extract groups since ICI 182,780 binds to the ER receptor and prevents the estrogenic extract from binding to it. See page 10 lines 346-349 for More details.

what dosages did use?

The concentration of ICI182,780 was 10 nM. This concentration does not cause cytotoxicity.

Point 3: this reviewer ask previously about mechanisms to this phenomenas. because this article suggested only effects of ICI 182,780-mediated inhibition of estogenecity of materials in MCF-7 cell model. Please suggest the identities of the estrogenic effects of the materials in MCF-7 cells.

Response 3: The focus of this research was not on the molecular mechanism.

In this study, we used ICI 182 780 to block ER receptor of the MCF-7 cells in order to determine whether the increased in cell proliferation is mediated by the estrogen receptor. Since it has been known that ICI 182,780 blocks estrogenic compounds from binding to the ER receptor. See Page 9 lines 300-305 for more details on the ICI 182,780 mode of action.

Reviewer 2 Report

I have no further comment.

Author Response

Reviewer 2 has no comment to address